# Alcohol policies in India: A scoping review

Jaclyn Schess[1,2], Lydia Bennett-Li[1], Richard Velleman[1,3]*, Urvita Bhatia[1,4], Alexander Catalano[1], Abhijeet Jambhale[1], Abhijit Nadkarni[1,5]

**1** Addictions Research Group, Sangath, Porvorim, Goa, India, **2** Department of Health Policy and Management, University of California, Berkeley School of Public Health, Berkeley, California, United States of America, **3** Department of Psychology, University of Bath, Bath, England, United Kingdom, **4** Oxford Brookes University, Oxford, England, United Kingdom, **5** Centre for Global Mental Health, Department of Population Health, London School of Hygiene & Tropical Medicine, London, England, United Kingdom

\* hssrdbv@bath.ac.uk

## Abstract

Globally, alcohol consumption causes significant societal harm and is a leading risk factor for death and disability in adults. In India, 3.7% of all deaths and 3.1% disability adjusted life years (DALYs) can be attributed to alcohol. In the context of rapid economic development and emphasized by the COVID-19 pandemic, India's lack of a consolidated and comprehensive alcohol policy has posed significant challenges to addressing this harm. In this context, the aim of our review was to undertake a comprehensive mapping of the State and national policy environment surrounding alcohol and its use in India, based on an analysis of policy documents. We did this though a scoping review of academic and grey literature, which helped to iteratively identify the websites of 15 international organizations, 21 Indian non-governmental organizations, and eight Indian Federal governmental organizations as well as State/Union Territory government sites, to search for relevant policy documents. We identified 19 Federal policy documents and 36 State level policy documents within which we have identified the specific policy measures which address the 10 categories of the World Health Organization's Global Action Plan to Reduce the Harmful Use of Alcohol. We found that there are major gaps in regulation of marketing and price controls, with much of this controlled by the States. In addition, regulation of availability of alcohol varies widely throughout the country, which is also a policy area controlled locally by States. Through the clear elucidation of the current policy environment surrounding alcohol in India, policy makers, researchers and advocates can create a clearer roadmap for future reform.

## Introduction

Globally, alcohol consumption is the cause of significant societal harm and is a leading risk factor for death and disability for adults. In 2016, 2.8 million deaths and 4.2% of the global burden of disability (Disability Adjusted Life Years- DALYs) were attributable to alcohol use [1]. Alcohol use disorders (AUDs), characterized by compulsive, chronic and heavy drinking despite harmful effects on health and relationships [2], affect 8.6% of men and 1.7% of women worldwide [3] and contributes 2.17% of total DALYs [4].

**Data Availability Statement:** All relevant data are within the manuscript and its Supporting Information files.

**Funding:** The authors received no specific funding for this work.

**Competing interests:** The authors have declared that no competing interests exist.

Global trends of alcohol use vary widely by region, with more consumption occurring in higher income countries [1]. However, as alcohol companies see developing countries as emerging markets for their products, alcohol availability and consumption continues to increase in low- and middle- income countries (LMICs) [5]. This is evident in India, where consumption has been steadily rising, with recorded per capita alcohol consumption increasing from 1.6 litres in 2003–2005 to 2.2 litres in 2010 [6] to 5.5 litres in 2016–2018 [7]. In India, although abstinence is high, almost one in five current drinkers has alcohol dependence [8]. In addition, 3.7% of all deaths and 3.1% of disability in India may be attributed to alcohol [4].

Importantly, even though consumption is higher in high-income contexts, alcohol use has a disproportionate effect on LMICs, with more than 85% of alcohol-attributable mortality occurring in these countries [9]. In addition, the poorest in any society are most likely to experience the harms of alcohol for a given amount and pattern of alcohol use [5, 10]; and this is especially relevant in LMICs where a larger proportion of the population still live in poverty.

Governments can mitigate the harms of alcohol use through an intersectoral public health approach. Such approaches include restrictions on availability and marketing of alcohol, higher taxes on alcohol, enforcement of drink-driving laws, and brief psychosocial treatments for AUDs [11]. These are both cost-effective and have differential impacts on poorer drinkers and therefore can counteract the health inequity inherent to the burden of alcohol-related harms [10].

Alcohol policy environments vary significantly across the world—67% of high-income countries, 43% of middle-income countries and only 15% of low-income countries have national alcohol policies [10]. Most of these policies do not place any restrictions on alcohol advertising and marketing, with "smaller countries, globally, and countries in Africa and the Americas most likely to have no restrictions" [12]. Even where restrictions are in place, alcohol companies quickly adapt to those restrictions with digital and other non-traditional marketing methods such as the use of online and social media approaches, new approaches in branding, and the utilization of marketing opportunities via branded events and products [13, 14]. In addition, treatment coverage for AUDs is generally low across the world, and very much lower in LMICs [10].

Most countries do levy some sort of tax on alcohol, though many do not adjust these for inflation nor use other pricing strategies. Despite implementing the "best buy" policy of alcohol availability restrictions, indicators show that availability is in fact increasing, particularly in low-income countries.

India comprises 36 States and Union Territories (UTs), and follows a federal republic system, where some policies are controlled at the Federal level and others are left to the States and UTs. In the case of alcohol, except for a handful of Federal policies (e.g. drink driving laws and health warning labels), most have been devolved to the States, resulting in alcohol control strategies looking entirely different even in close neighboring States. At the Federal level, India still lacks a comprehensive national policy on alcohol, but instead has focused on promoting prohibition-centered supply reduction and tertiary prevention, both implemented inconsistently and inefficiently [15]. States are responsible for drafting and implementing their own alcohol policies including setting a legal drinking age, place of sale restrictions, and excise taxes on alcohol products [10].

The inefficiency of the current alcohol policy environment has been particularly evident during the COVID-19 pandemic. India's attempt to control the virus involved a severe lockdown in 2020, which included a nationwide ban on alcohol sales [16]. Still, demand for alcohol remained, with Google Trends data showing that searches relating to alcohol withdrawal, how to extract alcohol from sanitizer, and alcohol delivery increased significantly during the lockdown [17]. In addition, enforcement of this prohibition was limited by both production of

illicit alcohol and smuggling; and alcohol remained available in several States even during this nationwide ban on alcohol sales, though at a premium [18]. Finally, before the pandemic, nine out of 10 with AUDs were not receiving treatment and the pandemic-related lockdown forced many into withdrawal with no support [16]. The COVID-19 pandemic has thus exposed critical gaps in public health policy and these call for renewed attention to be paid to health systems strengthening. It is therefore a critical time to evaluate the alcohol policy environment that exists in India.

One recent review [19] based on published and unpublished literature and anecdotal media information, concluded that the prevailing alcohol control policies and programs in India have been ineffective in controlling the burden of alcohol use and its associated impact. Yet currently there is no existing comprehensive review of alcohol policy documents in India, making it more difficult for researchers and policy makers to make intersectoral evidence-based reforms. In this context, we have undertaken a comprehensive mapping of the State and national policy environment surrounding alcohol and its use in India, based on an analysis of policy documents. We sought to answer the following research questions: (a)What current policies in India regulate the demand for and supply of alcohol and alcohol products? (b) What current policies impact on a range of societal outcomes arising from access to, use and misuse of alcohol? Our review synthesizing the landscape of alcohol policies throughout the country is the first such synthesis from India and will help policy makers and researchers to work towards building better responses to the growing alcohol use epidemic.

## Methods

The methods for this review follow the scoping review approach [20, 21]. Compared to a systematic review, the research question and search design in a scoping review are adapted iteratively to the knowledge which is developed throughout the search process [21]. Given how much is not known about the topic of interest and the varied sources of knowledge that needed to be examined, a scoping review with relevant adaptations was most suitable for the flexibility required. Our published protocol provides details of the approach [22] and we summarize it below and in Fig 1.

Our overall approach was a two-tiered, iterative search of literature, as depicted in Fig 1 and detailed as follows:

Tier 1 was an "academic search", in line with a standard systematic review. JS performed the search. JS and AJ performed screening and AN was consulted for conflict resolution. JS and AC performed synthesis. This search identified peer-reviewed literature about alcohol policies in India and this literature was used to identify further sources, particularly policy documents. We searched the following databases: MEDLINE, PsycINFO, Embase, Global Health,

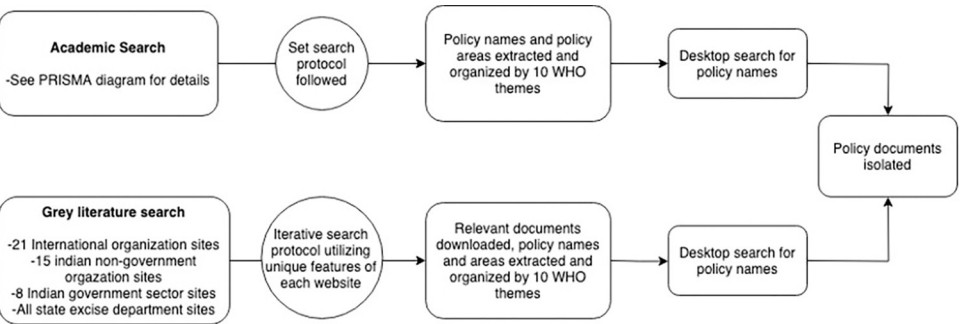

**Fig 1. Search process.** Flow chart depicting the process through which we identified policy documents.

and IndMed in August and September 2017. (Although the "policy search" described in the next paragraph was repeated in 2022, we decided not to update this "academic search", because the sole reason for doing that search was to identify policies, and we considered that the updating of the policy search described below would reveal any new or revised policies.)

Search terms focused on the themes of 'alcohol' (e.g. beer, wine, liquor, consumption, misuse, and synonyms), 'policy' (e.g. marketing, sales, production, importation) and 'India' (e.g. Indian Union, all States and UTs including historical names). All English language texts were included if they discussed policies related to alcohol or its impacts in India or its State/UTs. The following data was extracted from included articles: policy name (if mentioned), policy description, implementation date, and jurisdiction. The data was then collated according to the 10 policy categories outlined in the World Health Organization Global Strategy to Reduce the Use of Alcohol [23]. At this point, a desktop search was conducted to attempt to locate those policies that had been mentioned in the peer-reviewed articles.

Tier 2 was a "policy search" where we searched grey literature sources and government websites to identify policy documents (all government websites and all policy documents in India are published in English). A total of 21 international organization sites (e.g. World Health Organization, World Bank, Amnesty International), 15 Indian non-governmental sector sites (e.g. National Institute of Mental Health and Neuro-Sciences [NIMHANS], All India Institute of Medical Sciences [AIIMS]), and eight Indian governmental sector sites (e.g. Ministry of Social Justice & Empowerment, Ministry of Health and Family Welfare, and the India Code–an aggregator of all enforced Central and State Acts) were searched in March through May 2019 and again in January 2022. When accessing these sites, search bars were utilized to find the terms 'alcohol' or 'liquor' on sites for organizations based in India, or 'India, alcohol' on international sites. If search bars were unavailable on the site, a hand search was conducted, investigating different sections of the website applicable to Indian alcohol policy. Relevant articles were downloaded, and data was extracted in the same fashion as for the peer-reviewed articles.

Utilizing information gathered iteratively from the academic and grey literature, relevant policies were isolated and downloaded. Data was extracted including policy name, policy description, date implemented, and implementing jurisdiction. The main source of policy documents was the State/UT excise acts, obtained by visiting websites of individual State excise departments or equivalent. The most recent excise acts were downloaded, and detailed data was extracted (e.g. price controls, minimum age, point of sale restrictions, advertising restrictions, and more). In a small number of cases, we were unable to access State government websites, and hence were unable to access State excise acts. We did not pursue other means of obtaining these policy documents as the task at hand could be achieved with the large majority of State Excise Acts we had obtained. Further, among the excise acts we were able to access, some uncertainty remained as to their reliability due to issues of language, lack of textual clarity, missing sections and in some cases, the absence of the latest versions. Given these barriers, while we have attempted to extract reliable information from each State excise act, we have specified where these data issues created either absence of data or lack of clarity for our data extraction and have provided citations to the most comprehensive excise act documents available, and stakeholders interested in these policies in the future should seek out possible amendments to the cited excise acts.

This process allowed for synthesis of the general regulatory framework at the Federal and State level according to the 10 categories of the WHO Global Strategy. These 10 categories are: (1) leadership, awareness and commitment; (2) health services' response; (3) community action; (4) drink-driving policies and countermeasures; (5) availability of alcohol; (6) marketing of alcoholic beverages; (7) pricing policies; (8) reducing the negative consequences of

drinking and alcohol intoxication; (9) reducing the public health impact of illicit alcohol and informally produced alcohol; (10) monitoring and surveillance. Federal level policies and schemes, as well as State level policies, were synthesized according to these 10 categories.

## Results

Our academic search identified 47 relevant documents to review. The process for identifying these documents is laid out in the PRISMA diagram in Fig 2 [24]. The outcome of our academic search, describing the policies detailed and collated into the 10 categories outlined in the WHO Global Strategy, can be found in S1 Table. Informed by these findings and those of our grey literature search, as detailed in Fig 1, we identified 19 policy documents at the Federal level and 36 policy documents at the State level. This included 14 national policies, five national schemes, three State policies, and 33 State/Union Territory excise acts. This list excludes three States/Union Territories where the excise act was unavailable. All policy documents were obtained in the 2019 search, and eighteen updated versions were found in 2022. Most updated versions did not make substantial changes in areas relevant for this paper's

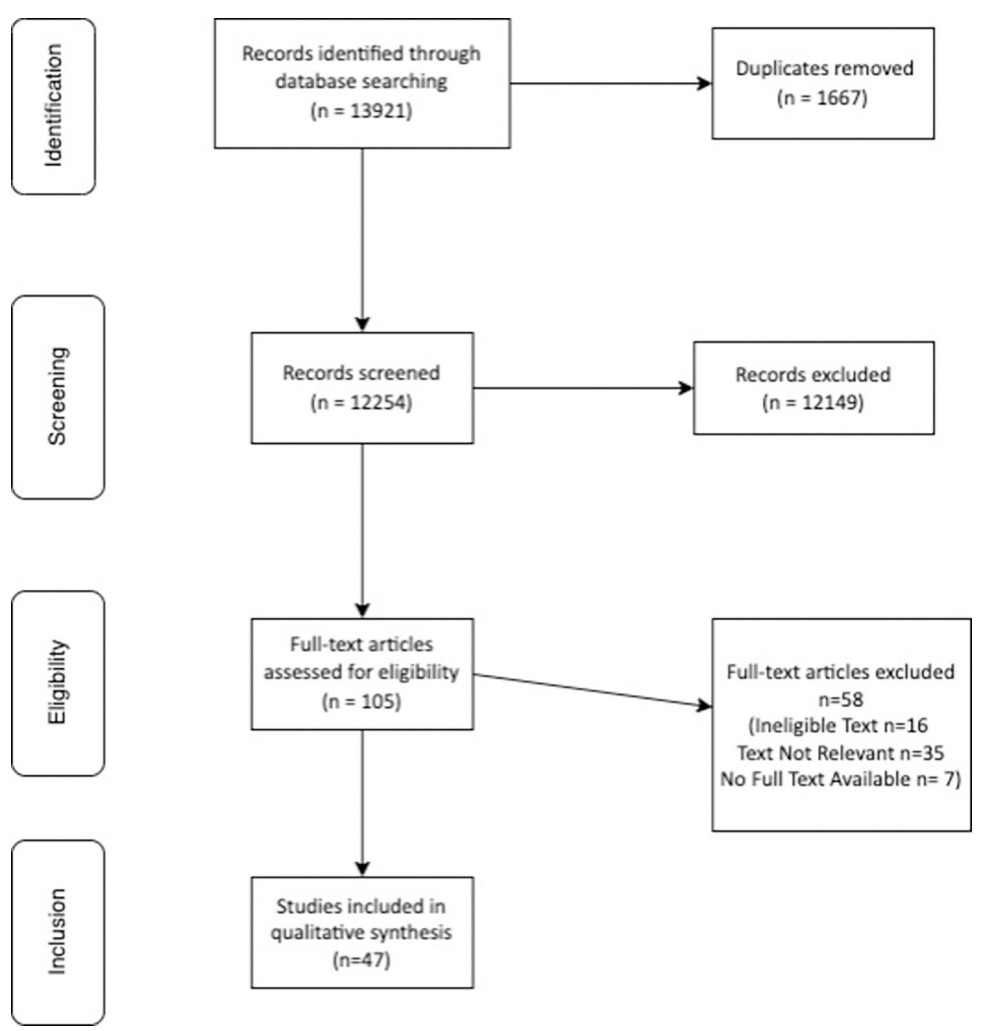

**Fig 2. Academic search PRISMA diagram.**

analysis. These policies were also collated according to the WHO Global Strategy and this is included in S2 Table with State excise acts listed separately.

Alcohol policies in India are implemented with a mix of Federal and State level policies. Some are Federal "Schemes" which are national directives, implemented by the States. Most of the alcohol-related policy implemented at the State level is written in excise policies, which cover each State's individual policies regarding taxes, manufacture, transport, import, labeling, sale, and other regulation of liquor. In comparison, Federal level policies are usually more specific in their nature, addressing a single aspect, such as drink driving, journalistic norms, or prevention of AUDs, in an individual policy.

## Leadership, awareness and commitment

We identified two national schemes which address the harmful use of alcohol, but no central policy covering this issue. For example, the Central Sector Scheme of Assistance for Prevention of Alcoholism and Substance (Drugs) Abuse and For Social Defense Services funds education on alcohol use, but it does not detail a coordinated national approach to awareness raising [25]. In addition, the Mental Health Care Act of 2017, grants rights to those with mental ill health, including drug and alcohol dependence, to mental healthcare and includes significant changes in the previous law (Mental Health Act of 1987) calling for coordination across sectors to implement these rights [26].

## Health services' response

We identified six national level policies. There were schemes delineating the health services' response to AUDs. For example, the Central Sector Scheme of Assistance for Prevention of Alcoholism and Substance (Drugs) Abuse and For Social Defense Services provides funding to non-government organizations which provide de-addiction services [25].

In addition, national level policies were identified which target health service provision. The Drug Deaddiction Program provided one-time grants to establish 122 de-addiction centers associated with district hospitals and psychiatry departments with the All India Institute of Medical Sciences (AIIMS) as the coordinating body [27] and the provision of services was improved with an updated scheme to enhance the de-addiction centres and establish Drug Treatment Centres [28]. The Indian Public Health Standards, Guidelines for District Hospitals provide guidelines for treatment and follow up of AUDs [29]. The establishment of the Health and Wellness Centres through the Ayushman Bharat universal healthcare scheme includes mental health and substance use treatment in decentralized health centers, though policy documents provide guidelines on this provision not policy mandates [30].

We identified only one state-level policy, the Punjab Substance Use Disorder Treatment and Counseling and Rehabilitation Centres Rules of 2011, which recommended specific standards for substance use disorder treatments in the State [31].

## Community action

The only policy we found which implies supporting community action was The Provisions Of The Panchayats (Extension To The Scheduled Areas) Act of 1996, which specifically endows the Scheduled Areas panchayats (village governance bodies) pan India with the power to enforce regulations or restrictions on the sale or consumption of alcohol [32].

## Drink driving policies and countermeasures

Two relevant policies were identified. The Motor Vehicles Act of 1988 specifies the legal drinking limit as a blood alcohol level of 30 mg per 100 ml and authorizes uniformed officers to

routinely conduct breathalyzer tests on anyone they reasonably suspect to be drink driving [33]. There was also a ban on alcohol sales within 500 meters of a National or State highways passed in 2017 [34].

In addition, the Federal government has focused on trying to limit the consequences of road accidents, as they are frequently related to alcohol consumption. The Scheme 'Capacity Building for developing Trauma Facilities on National Highways' aims to decrease preventable death from road accidents by upgrading and finishing pending healthcare facilities to respond to trauma incidents [35].

## Availability of alcohol

In general, supply reduction is taken up by the States, where excise acts focus the majority of their detailed regulations on this approach. States vary in their supply reduction approaches: from total prohibition through to State manufacture and distribution of alcohol. In addition, most excise acts include detailed regulations regarding the sale and consumption of liquor. It is important to note that the 1996 Provisions of the Panchayats Act specifically empowers India's tribal regions to enact their own alcohol control policies specifically with regard to prohibition, and/or restricting the sale and consumption of intoxicants, irrespective of which State they are in and the control approach taken by that State [32].

**Total prohibition.** Presently, five States, namely Bihar, Gujarat, Mizoram, parts of Manipur, and Nagaland, completely prohibit the sale and consumption of alcohol (e.g. have prohibition in place, or are 'dry' States) [36–41].

**Licensing and manufacture.** Except those five where import, export, transport and manufacture are banned by prohibition, all States/UTs regulate the import and export of alcohol. Further, all others include in their excise acts, regulations for alcohol's transport, and require licensing for its manufacture and sale [41–64].

Certain States go further than this, having centralized systems of distribution of alcohol to regulate supply. Evidence of a State or government-controlled distribution system for alcohol was found in six States [38, 48, 49, 65, 66]; however, there was some lack of clarity as to the extent of government control over some of these distribution systems. For example, the state of Tamil Nadu maintains a State-run monopoly on alcohol wholesale and retail sales through the Tamil Nadu State Marketing Corporation [65].

**Sale and consumption restrictions.** Of the 33 States/UTs analyzed, 18 ban the sale and/or consumption of alcohol in public places [37, 38, 40–42, 44, 46, 47, 55, 56, 58, 61, 63, 66–70], leaving 14 without such regulation, and one State (West Bengal) where the excise act was unclear. Regulations controlling the days and hours of sale of alcohol were present in the excise acts of 26 States/Territories [41–46, 48–50, 53–55, 57–61, 63, 65–68, 71–75]. The remaining seven States/Territories had no evidence of such regulation.

Specific policies apply to special populations such as foreign individuals staying in hotels in States with prohibition where, for example, some of these States allow consumption of liquor within certain classes of hotel. In addition, liquor is banned within military cantonments except for military officers [76].

The minimum age for alcohol purchase and consumption varies across States/UTs, being 18 years in 11 States/Territories [44, 47, 50, 53, 55, 56, 58, 60, 62, 75, 77], 21 years in 13 States/Territories [43, 45, 46, 48, 49, 54, 57, 63–66, 73, 74], 25 years in two States/Territories [67, 68] and 26 years in one Territory [61]. There were an additional five States/Territories where the excise act did not specify this point.

Further, 24 of the 33 excise acts included a limit for the maximum retail sale to individuals and/or individual alcohol possession limit [38, 42–50, 54–56, 61–64, 66, 67, 71, 73–75]. The

extent of this regulation varied significantly between excise acts. For example, the Goa Excise Act refers to a maximum amount of alcohol a person may have in their possession (but does not specify this amount) [46]. The Kerala Excise Act regulates only the maximum amount of alcohol allowed to be transported by an individual [75]. The remaining nine excise acts had no evidence of such regulation.

The location of alcohol sale outlets was regulated by various means, including 'distance from' certain sites such as schools or religious institutions, and by regulating alcohol outlet density. Across the excise acts, the location of alcohol sale outlets was predominantly regulated by controlling their distance from certain sites. Only seven of the 33 excise acts did not include any 'distance from' regulations. Alcohol outlet density regulation (limiting the number of alcohol outlets permissible within a certain distance from each other) was found to varying degrees, in 14 of the 33 excise acts [43–45, 47, 48, 50, 54, 56, 57, 59, 60, 63, 65, 71, 73]. For example, in the Haryana Excise Act, there is substantive detail prohibiting the sale of alcohol within 150 metres from a college, school, bus stand or place of worship, as well as in view of a national or State highway [63]. Similarly, the Goan Excise Act prohibits alcohol outlets within 100m from specific locations like educational institutions and places of worship [46]. In total, five States/UTs were found to have no evidence of location of alcohol sale outlet regulation.

### Marketing of alcoholic beverages

There were three national codes identified that specify that alcohol advertisements cannot be featured in written media or on the radio. On TV, the 2009 Amendment to Cable Television Network Rules specifies that cable advertisements cannot refer to alcohol products, but their brand-names can still be advertised [78].

State/UT excise acts also include regulation of the advertisement of alcohol products. A ban on the advertisement, promotion or sponsorship of alcohol products was found in the excise acts of all but 10 States [37–41, 43–46, 55, 57, 58, 61–63, 65, 67, 68, 71]. Further, a ban on advertising at the point of alcohol sale was found in the excise acts of 21 States, leaving 12 without such regulation [38–41, 43–46, 56–58, 61–63, 65, 67, 68, 71, 73].

### Pricing policies

The Federal Goods and Services Tax (GST) Act specifies that no sales tax is to be levied on liquor [79]. Taxes on liquor are instead implemented at the State level, detailed in all excise policies. The level of detail regarding taxation across the excise policies was overall relatively vague. For example, the Maharashtra excise act mentions the existence of a tax on liquor items, recognized as the 'excise revenue', but does not list any further detail such as the percentage or a range of percentages to be applied [71]. Instead, all excise acts disclose that the State government has the authority to make and carry out rules in relation to excise revenue. The Kerala excise act has attempted to include a maximum amount of tax which can be applied per proof liter of Indian made alcohol, however there have been several revisions to this element of the act, making it somewhat unclear as to which figure is currently in application [75].

The regulation of alcohol pricing was found in 19 of the 33 States/UTs, including the five with prohibition in place. Of the 19 with alcohol pricing regulation, nine States/UTs regulated only the maximum price of alcohol [46, 58, 62, 64–66, 71], with the remaining 10 regulating both the maximum and minimum price [44, 53, 61, 63, 68, 74].

### Reducing the negative consequences of drinking and alcohol intoxication

Four national policies were identified at the national level. Health warnings exist at the Federal and State level to mitigate the negative consequences of drinking.

The Food Safety and Standards (Alcoholic Beverages) Regulations of 2018 is a Federal effort to create standard requirements for ingredients in alcoholic beverages and maintain labeling requirements for these items. All alcohol packages must specify the alcohol content in percent alcohol by volume or proof. Further, the packaging must specify the number of 'standard drinks' contained within the package. In addition to this, the regulation clarifies that alcoholic beverages must not contain wording that would imply that it is non-intoxicating, non-alcoholic, or that it includes any health benefits to the user. Additionally, all alcohol containers must contain a sign no smaller than 3mm in English or the language of the States' own choice, indicating that alcohol is injurious to health, and warning the user to not drink and drive [80].

At the State/UT level, six of the 33 excise acts required warnings or security holograms on all alcohol bottles and other retail containers [45, 59–61, 63, 65, 66, 77, 81, 82]. However, the exact details of these requirements varied across these six excise acts. For example, in Odisha, the excise act requires alcohol labels to include the alcohol percentage and the health warning; 'Drinking liquor is injurious to health' [59, 60]. Whereas, in Tamil Nadu, the State excise act regulates only the use of polyester hologram excise labels as a security safeguard [81, 82].

## Reducing the public health impact of illicit alcohol and informally produced alcohol

Regulating illicit and informally produced alcohol is left to the States. In 17 States/UTs informally, locally produced alcohol known as country liquor or toddy is explicitly allowed in their policies but with some amount of availability regulation [42–44, 46–49, 51, 55–60, 62, 66, 68, 71]. Five states explicitly ban these informally produced liquors [39, 41, 52, 65, 74, 75]. All excise acts specify penalties for contravening the act, such as illegal production or sale of liquor.

## Monitoring and surveillance

One policy, the Central Sector Scheme of Assistance for Prevention of Alcoholism and Substance (Drugs) Abuse and For Social Defense Services has included the commissioning from the Ministry of Social Justice of data collection and reporting on the use of alcohol in India, but there is not a coordinated approach across all sectors where alcohol use might create harm [25].

## Discussion

We utilized a relatively underutilized search method to conduct a search on Indian alcohol policy, a topic that has yet to be covered by peer-reviewed literature in a manner that comprehensively approaches the finding of policy documents and the synthesizing of their provisions. Similarly to other studies of alcohol policy in India which have utilized a different methodology [19], we find a fractured approach which leads to widely different policy directives across the country. Uniquely, this review directly cites the existing policies in order to elucidate the current policy environment, allowing researchers, policy makers and planners to set future priorities for better regulation of alcohol products, and the prevention and treatment of AUDs in India, as well as other developing countries which are struggling with similar emerging problems related to alcohol use, as alcohol companies look for new markets.

The national policy landscape elucidated in this review identifies key elements of the WHO Global Strategy which have great potential to reduce harm. These include national health provision schemes such as the Drug Deaddiction Program, a national blood alcohol content limit for driving, as well as renewed attention to rights for those in the mental health system through the Mental Health Act. Still, this review shows in detail that the current alcohol policy environment is insufficient to mitigate the harms of rising alcohol consumption in India. While the

country does have some national policies in line with the World Health Organization Global Strategy to Reduce the Harmful Use of Alcohol, much of the responsibility for curbing harmful alcohol use is left to the States and Territories. This means that cost-effective measures like pricing adjustments, advertising bans, regulation of place and hours of sale, and minimum age for consumption are included in State policies to largely varying degrees across the country.

As outlined in the WHO Global Strategy to Reduce the Harmful Use of Alcohol, there are several effective policies supported by global evidence to reduce harmful alcohol use and its consequences. Three of these policy areas are identified as the most effective and cost-effective interventions to reduce alcohol related harm: strong restrictions on alcohol availability, bans or comprehensive restrictions on alcohol advertising across platforms, and strategic increases of alcohol excise taxes [12]. This review highlights substantial gaps in India in the application of these evidence-based policy interventions. Regarding alcohol availability: this policy matter is handled by the States and varies widely, from complete prohibition; to restrictions on time and place of sale; to very little regulation of availability. Regarding restrictions on advertising, these exist on traditional media like newspaper and radio, though loopholes exist for television and there is no comprehensive approach to new media. Certain States also have bans or restrictions on advertisements, including point of sale advertisements, but many States do not have these restrictions. Finally, as will be discussed, excise taxes have been increased, but frequently not with the interests of public health in mind, leaving much to be desired for alcohol pricing policy.

The Indian context therefore highlights an example of a hybrid model where the Federal and State governments share the responsibilities for both developing and implementing alcohol policy. There are of course hybrid models where the Federal level takes on coordination, and States then adapt and implement that policy; but this is not the model adopted in India. Within Indian alcohol policy, lack of Federal coordination leads to distinct differences in policy environments across the country which means that an individual's experience with alcohol use and related harm is likely to be largely influenced by their location.

In some ways, the hybrid model lends benefits, as States can respond to the needs of their populations. We see this in Punjab, for example, where the State government has implemented a targeted substance use policy in response to an opiate crisis; at 0.8% [83], Punjab has the second highest population rate of people with opioid use disorder in the country [8]. It also allows for States to experiment with different policy approaches. For example, other States (including Andhra Pradesh, Haryana, Kerala, Tamil Nadu) have mandated prohibition in the past but have repealed those measures. Reasons for abandoning the policy included difficulties with enforcement in the face of smuggling across State borders and illegal sales of informally produced alcohol [84] and sharp decreases in tax revenues from excise tax on alcohol [15].

This highlights one of the main conflicts that exist for State governments in managing alcohol policy under a hybrid approach. In the current set-up, States face a conflict of interest: they are responsible for providing healthcare and protecting public health, while also receiving large percentages of their tax revenue from excise taxes on alcohol. In the 2019–20 Fiscal Year, State/UT alcohol excise revenue totaled 2.25 trillion rupees (approximately 30 billion USD). Twenty-one States made more than 15% of their yearly revenue from alcohol excise tax [85]. In practice, financial considerations take priority over health concerns. This calculation came into play clearly when States decided to repeal prohibition and was again highlighted during the COVID-19 pandemic. As India implemented its first nationwide lockdown in March of 2020, liquor shops were shut down across the country, forcing many into withdrawal, creating a black market for liquor and having other unintended consequences [16]. When reopening liquor stores, it was clear that the loss of tax revenues and industry pressure took priority, as COVID-19 guidelines took a backseat to re-opening liquor sales [86, 87].

What is clear though is that the hybrid model means that there are significant variations in State policy, and that these lead to widely different public health outcomes: for example, the prevalence of current drinking amongst males varies from 62.1% in Tripura, where there is a significant amount of informal alcohol produced and consumed, to 1.7% in Bihar, although here (as in other States where prohibition is in place) there is likely to be significant underreporting [8].

One of the major issues that arose in this examination of existing policies is the lack of clarity around many of the provisions, even if they are mentioned. For example, although all States have policies related to taxes on alcohol, there is both great variability across these taxes, as well as a lack of clarity about both what percentage or amount might be levied, and how that amount might change with respect to inflation. The specification of the state's ability to tax instead of adding these details regarding taxation within the excise acts is likely to be intentional, as it allows State Governments to change tax rates without the need to frequently update their excise acts. Similarly, although several States have policies which lay down the maximum price that alcohol could be sold at, only 10 lay down a minimum price. Evidence from countries which have utilized a minimum unit price for alcohol has shown that this is an important tool with respect to reducing alcohol-related harm [88]. In addition, there are policies which remain rife with loopholes that are taken advantage of by alcohol companies. For example, the Cable Television Network Rules allows for alcohol company brand names to be used in advertising. There has therefore been a proliferation of "surrogate advertising": adverts which promote a non-alcohol product, i.e. bottled water or a cola, but which are branded with an alcohol company's branding and frequently using imagery associated with alcohol [15]. This has had a particularly strong impact on young people [89], with advertising and marketing including through sports sponsorship, influencer marketing, and new media influencing the likelihood that young people drink and drink heavily [90]. The advertising industry self-regulates through the Advertising Standards Council of India which put out new rules in 2020–2021. So far, the ASCI appears to have taken a more keen interest in limiting surrogate advertising [91, 92], but it is yet to be seen how this will impact advertising practice over time.

Even where there are clear policies in place, there are major questions as to the implementation and enforcement of these regulations. This review has focused on locating and describing existing policies in place, but this does not tell us to what extent these policies are in fact being implemented and enforced as they are written: clearly differential enforcement would create even further disparity between States on alcohol control measures. Previous studies in India have indicated poor or disparate enforcement of a range of alcohol policies–examples include minimum drinking age and drink driving laws, and the prevalence of illicit liquor production and trade [15, 93, 94]. In addition, smuggling and other enforcement weaknesses are prevalent in States that have prohibition laws [95–97]. Policies such as the Mental Health Act of 2017 faces clear implementation challenges as it is not accompanied by an increase in resources to deliver care that is called for in the act. The focus of this work also means that there may be activities conducted by the Government of India, such as monitoring excise related crimes, for which we could not identify as specific policy, but which are impacting use of alcohol or its societal impact [98]. A number of health system responses may be addressing alcohol use, such as the District Mental Health Programme [99] and the Narcotic Drug and Psychotropic Substance Act [27] and its implementation plan [28], but they do not identify alcohol use as a focus of the policy in the text and so have not been included in the results but may still be relevant for outcomes. Further work that systematically assesses the extent to which alcohol policies are implemented and enforced, in addition to eliciting the relevance of policies which are not targeted at alcohol use but may be implemented such that those with alcohol use are included, would be a critical next step to understanding how to translate policy to improve public health and reduce alcohol-related harms.

There are a few limitations of this study to note. First, we only looked at English language texts in the academic search, which may have limited our initial findings. This has been handled by performing a wide grey literature search and performing translation of texts in Hindi where possible. In addition, some policies, particularly a handful of State excise acts, were not accessible. This was only the case for a small percentage of the States, and we therefore believe this does not have a large impact on our findings. While we cannot therefore ensure that we have found every single policy that applies to the alcohol policy environment, the combination of academic and grey literature searches conducted in a systematic manner made use of the scoping review method in a way that we believe had highest likelihood of finding relevant policies.

COVID-19 has highlighted the need for health system planners to think seriously about the burden of mental health and substance use disorders. While it is too early to tell the full impact of the pandemic on population mental health, it is clear that the bereavement, uncertainty, job loss and isolation associated with COVID-19 are having significant mental health consequences in India [100, 101] and around the world [102], with many people also experiencing increased alcohol use or significant alcohol withdrawal depending on their access to alcohol [16, 103]. There was already significant unmet need for mental health care, particularly in low-resource settings. Now, 93% of countries are reporting disruption of their mental, neurological and substance use disorder services [102]. Chronic underfunding of the mental health sector means that health services are ill equipped to handle the increasing demand for services. This is particularly true for AUDs in India, which have the highest treatment gap in the country of all mental and substance use disorders [104]. It is imperative that health system rebuilding from COVID-19 includes investment in prevention and treatment of AUDs.

In conclusion, our systematic policy review maps the current policy environment around alcohol in India and in particular highlights the large variations which exists across the country. The lack of coordination inherent in the type of hybrid Federal-State policy system used in India leads to widely different experiences across the country on most policy areas with respect to alcohol. In the wake of the COVID-19 pandemic, the Indian Federal government has an opportunity to respond to the rising mental health needs of its population by implementing a public-health informed national response to alcohol use. Key recommendations for policy-makers that emerge from this analysis are as follows: (1) A national alcohol policy would reduce the harms from alcohol while improving equity across the country; (2) Implementation of price controls and availability regulations, known to be effective and cost-effective at the federal level have potential to have the most significant impact; (3) the prevention and treatment of alcohol use disorders must be prioritized on the state and national level in health system rebuilding in order to reduce the impact of alcohol related harms.

## Supporting information

**S1 Table. Data obtained from academic search.** Outlines policy details obtained during the academic search stage of the review.
(DOCX)

**S2 Table. Policy document results.** Outlines the details of the policy documents obtained through all stages of the search at both Federal and State/Territory levels.
(DOCX)

## Author Contributions

**Conceptualization:** Jaclyn Schess, Richard Velleman, Urvita Bhatia, Abhijit Nadkarni.

**Data curation:** Jaclyn Schess, Lydia Bennett-Li.

**Formal analysis:** Jaclyn Schess, Lydia Bennett-Li.

**Investigation:** Jaclyn Schess, Lydia Bennett-Li, Alexander Catalano, Abhijeet Jambhale.

**Methodology:** Jaclyn Schess, Lydia Bennett-Li, Richard Velleman, Urvita Bhatia, Abhijit Nadkarni.

**Project administration:** Jaclyn Schess.

**Supervision:** Abhijit Nadkarni.

**Validation:** Richard Velleman, Abhijit Nadkarni.

**Visualization:** Jaclyn Schess.

**Writing – original draft:** Jaclyn Schess.

**Writing – review & editing:** Jaclyn Schess, Lydia Bennett-Li, Richard Velleman, Urvita Bhatia, Alexander Catalano, Abhijit Nadkarni.

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
