## [Decision Letter · Decision Letter 0]

15 Aug 2023

PONE-D-22-23985Alcohol policies in India: a scoping reviewPLOS ONE

Dear Dr. Velleman,

Thank you for submitting your manuscript to PLOS ONE. After careful consideration, we feel that it has merit but does not fully meet PLOS ONE’s publication criteria as it currently stands. Therefore, we invite you to submit a revised version of the manuscript that addresses the points raised during the review process.

Your manuscript has been evaluated by three reviewers, and their comments are appended below. The reviewers have identified the methodology reported in this study as needing elaboration regarding details of the protocol. In particular, Reviewer #3 has questioned the end date of the literature search in the context of the objective to assess the impact on societal outcomes. Please ensure you address each of the reviewers' comments when revising your manuscript.

We look forward to receiving your revised manuscript.

Kind regards,

Hugh Cowley

Staff Editor

PLOS ONE

Journal Requirements:

Additional Editor Comments (if provided):

The manuscript can be accepted

Reviewers' comments:

Reviewer's Responses to Questions

**Comments to the Author**

1. Is the manuscript technically sound, and do the data support the conclusions?

Reviewer #1: Yes

Reviewer #2: Yes

Reviewer #3: Yes

2. Has the statistical analysis been performed appropriately and rigorously? 

Reviewer #1: N/A

Reviewer #2: N/A

Reviewer #3: Yes

3. Have the authors made all data underlying the findings in their manuscript fully available?

Reviewer #1: Yes

Reviewer #2: Yes

Reviewer #3: Yes

4. Is the manuscript presented in an intelligible fashion and written in standard English?

Reviewer #1: Yes

Reviewer #2: Yes

Reviewer #3: Yes

5. Review Comments to the Author

Reviewer #1: 1. Scoping review methodology should be clearly stated including reporting checklists(guideline)

2. Eligiblity criteria for the selection of previously existed alcohol policies should be clearly stated.

3. You have to use the updated references, i.e reference 14,24,31 and 32

Reviewer #2: The review is comprehensive and methodology is sound. But the authors are suggested to cite references in the texts of results section in addition to S1 Table. It is also needed to check spelling at the line 174 for surveillance and confirm the phrase "abv or as proof" at line 340.

Reviewer #3: A critical policy review. This a reasonable effort from the authors, as covering a large amount of grey literature on the topic, is a tough job.

The major drawback is not repeating the academic search after 2017. In 6 years, as the authors pointed out and also mentioned the impact of COVID-19, there has been research on the impact of the lack of alcohol policy during COVID-19. Authors should consider repeating the academic search.Though you have stated, "because the sole reason for doing that search was to identify policies, and we considered that the updating of the policy search described below would reveal any new or revised policies" But, your objective also includes impact on societal outcomes, how will you know that societal outcomes just from policy documents?

1. Although the authors stated the objective of the impact of societal outcomes, I could only see a little information in the results or discussion in a structured way.

2. You found significant gaps. How did you find the gaps? What parameter is used? It is unclear to the reader how the authors concluded.

3. Where does the country stand on the alcohol policy index based on the author's assessment, is there a change that the authors noted based on the assessment from previous studies? Ref: https://doi.org/10.1371/journal.pmed.0040151

4. Leadership, awareness and commitment: What about NAPDDR 2018 (national action plan on drug demand reduction 2018), RPWD 2016 including mental illness (also substance use).

5. Health services’ response: What about DTC program, ATF, Ayushman bharath health and wellness centres, Nasha mukht Bharath Abayan?

6. Under community action, NAPDDR 2018 use CLPI community lead peer intervention https://grants-msje.gov.in/omrunningcpli

7. Drinking and driving poilcies and counter measures: Ban on alcohol sales 500m on national highways: https://morth.nic.in/sites/default/files/circulars_document/SR-2017.06.01-Ban%20on%20liquor%20shops%20along%20National%20Highways.pdf

8. Under illicit alcohol: what about unrecorded alcohol use and related policies? https://www.iard.org/getattachment/fdd90791-41cb-4bd3-98f0-555fbf9818f8/unrecorded-alcohol-in-india.pdf

9. Under monitoring and surveillance: Government of India monitors, regularly updates the excise related crime and enforcement process as a part of supply reduction https://ncrb.gov.in/sites/default/files/CII-2021/CII_2021Volume%201.pdf

10. Finally, the overall discussion appears towards the overall (hybrid) system, maybe gaps in comparison to global standards, deviation or less evidence informed policies needs to be depicted. Further, there is a significant amount of grey literature available, authors should be congratulated for thier efforts in covering such a vast literature, a final round of consultation with policy experts might improve further.

Best wishes.

6. PLOS authors have the option to publish the peer review history of their article (what does this mean?). If published, this will include your full peer review and any attached files.

Reviewer #1: **Yes: **Agmas Wassie Abate

Reviewer #2: No

Reviewer #3: **Yes: **Dr. Venkata Lakshmi Narasimha

---

## [Author Response · Author response to Decision Letter 0]

10 Oct 2023

We have submitted a 'Response to Reviewers' document that lists each comment and responds to it.

---

## [Decision Letter · Decision Letter 1]

2 Nov 2023

Alcohol policies in India: a scoping review

PONE-D-22-23985R1

Dear Dr.Vellemann

We’re pleased to inform you that your manuscript has been judged scientifically suitable for publication and will be formally accepted for publication once it meets all outstanding technical requirements.

Kind regards,

George Kuryan

Academic Editor

PLOS ONE

Additional Editor Comments (optional):

Reviewers' comments:

Reviewer's Responses to Questions

**Comments to the Author**

1. If the authors have adequately addressed your comments raised in a previous round of review and you feel that this manuscript is now acceptable for publication, you may indicate that here to bypass the “Comments to the Author” section, enter your conflict of interest statement in the “Confidential to Editor” section, and submit your "Accept" recommendation.

Reviewer #2: All comments have been addressed

Reviewer #3: All comments have been addressed

2. Is the manuscript technically sound, and do the data support the conclusions?

Reviewer #2: Yes

Reviewer #3: Yes

3. Has the statistical analysis been performed appropriately and rigorously? 

Reviewer #2: N/A

Reviewer #3: N/A

4. Have the authors made all data underlying the findings in their manuscript fully available?

Reviewer #2: Yes

Reviewer #3: Yes

5. Is the manuscript presented in an intelligible fashion and written in standard English?

Reviewer #2: Yes

Reviewer #3: Yes

6. Review Comments to the Author

Reviewer #2: Inclusion of citation in text is found in revised version and thanks for your efforts. Spelling errors are also corrected according to my previous comment. Although you mention as line 186 and 370, corrected words are found in line 176 and 352. There is no additional comment at this review for R1 and hope for successful publication.

Reviewer #3: The authors have addressed all the comments adequately.

Some suggestions for consideration

1. The focus of NAPDDR is not just the NDPS; it is also implemented in the background of the Central sector scheme for alcohol. Centres like IRCAs also cater to people with alcohol use disorders. As this paper potentially impacts policymakers and stakeholders involved in service delivery, a mention of NAPDDR might be useful.

2. RPWD talks about mental illness, as the definition of mental illness according to MHCA includes substance use, and RPWD includes substance use (although they explicitly don't mention about it).

3. ATF doesn't come under DDAP. It's under MOSJE. Similarly, there are differences in DTC and DAC. Mentioning these different measures from the state might be helpful.

7. PLOS authors have the option to publish the peer review history of their article (what does this mean?). If published, this will include your full peer review and any attached files.

Reviewer #2: No

Reviewer #3: **Yes: **Venkata Lakshmi Narasimha

---

## [Editor Report · Acceptance letter]

7 Nov 2023

PONE-D-22-23985R1 

Alcohol policies in India: a scoping review 

Dear Dr. Velleman:

I'm pleased to inform you that your manuscript has been deemed suitable for publication in PLOS ONE. Congratulations! Your manuscript is now with our production department. 

Kind regards, 

on behalf of

Professor George Kuryan 

Academic Editor

PLOS ONE